**Data Availability Statement:** Data cannot be shared publicly because of all data belong to the Navajo Nation. Data may be requested by

# Changes in food pricing and availability on the Navajo Nation following a 2% tax on unhealthy foods: The Healthy Diné Nation Act of 2014

Carmen George[1,2], Carolyn Bancroft[1,3], Shine Krystal Salt[1,2], Cameron S. Curley[1,2], Caleigh Curley[3], Hendrik Dirk de Heer[3], Del Yazzie[4], Regina Eddie[5], Ramona Antone-Nez[4], Sonya Sunhi Shin[1,2]*

1 Brigham and Women's Hospital, Boston, MA, United States of America, 2 Community Outreach and Patient Empowerment, Gallup, NM, United States of America, 3 Department of Health Sciences, Northern Arizona University, Flagstaff, AZ, United States of America, 4 Navajo Epidemiology Center, Window Rock, AZ, United States of America, 5 College of Nursing, Northern Arizona University, Flagstaff, AZ, United States of America

* sshin@bwh.harvard.edu

## Abstract

### Introduction

In 2014, the Navajo Nation Healthy Diné Nation Act (HDNA) was passed, combining a 2% tax on foods of 'minimal-to-no-nutritional value' and waiver of 5% sales tax on healthy foods, the first-ever such tax in the U.S. and globally among a sovereign tribal nation. The aim of this study was to measure changes in pricing and food availability in stores on the Navajo Nation following the implementation of the HDNA.

### Methods

Store observations were conducted in 2013 and 2019 using the Nutrition Environment Measurement Survey-Stores (NEMS-S) adapted for the Navajo Nation. Observations included store location, type, whether healthy foods or HDNA were promoted, and availability and pricing of fresh fruits and vegetables, canned items, beverages, water, snacks and traditional foods. Differences between 2013 and 2019 and by store type and location were tested.

### Results

The matched sample included 71 stores (51 in the Navajo Nation and 20 in border towns). In 2019, fresh produce was available in the majority of Navajo stores, with 71% selling at least 3 types of fruit and 65% selling at least 3 types of vegetables. Compared with border town convenience stores, Navajo convenience stores had greater availability of fresh vegetables and comparable availability of fresh fruit in 2019. The average cost per item of fresh fruit decreased by 13% in Navajo stores (from $0.88 to $0.76) and increased in border stores (from $0.63 to $0.73), resulting in comparable prices in Navajo and border stores in 2019.

contacting the Navajo Human Research Review Board (contact via mrwinney@navajo-nsn.gov). The authors collected the data but were also required to obtain NHRRB approval to do so; therefore, authors did not have special access to data, and all interested researchers may access the data by obtaining NHRRB approval, as the authors have.

**Funding:** National Institute On Minority Health And Disparities, National Institutes of Health R01MD013352 (DDH, SSS) https://www.nimhd.nih.gov National Institute On Minority Health And Disparities, National Institutes of Health U54MD012388 (DDH). https://www.nimhd.nih.gov The funders had no role in study design, data collection and analysis, decision to publish, or preparation of the manuscript.

**Competing interests:** The authors have declared that no competing interests exist.

While more Navajo stores offered mutton, blue corn and wild plants in 2019 compared to 2013, these changes were not statistically significant.

## Discussion

The findings suggest modest improvements in the Navajo store environment and high availability of fruits and vegetables. Navajo stores play an important role in the local food system and provide access to local, healthy foods for individuals living in this rural, tribal community.

## Introduction

The Navajo Nation is one of the largest tribal nations in the United States and worldwide, both in terms of enrollment and geographic area [1,2]. Traditionally, the Navajo people lived a healthy lifestyle characterized by physical activity and consumption of healthy, traditional foods. However, a combination of poverty [1], inadequate and discriminatory policies and disruptions in food systems [3] have led to high rates of food insecurity [4] and nutrition-related chronic diseases such as type 2 diabetes [5,6]. Currently, similar to many American Indian/Alaskan Native (AI/AN) people, the Navajo experience limited access to healthy foods.

Across the Navajo Nation, an area approximately the size of West Virginia, there are only 13 grocery stores, resulting in the area being labeled a 'food desert' [7]. There are a larger number of convenience stores, but prior research suggests that convenience stores on and around the Navajo Nation offer more processed foods with minimal nutritional value and at higher prices compared to grocery stores in border towns around the Navajo Nation [8,9]. Proximity, cost and product shelf-life have also been documented as key factors impacting Navajo families' healthy food purchasing [10,11]. Navajo store managers have demonstrated interest in offering healthier options, but barriers include limited fruit and vegetable choices from distributors and concerns about perceived low customer demand for healthier items [12].

To promote the health of the Navajo people and increase access to healthy foods, the Healthy Diné Nation Act (HDNA) was passed in November 2014, which included a 2% added surtax on unhealthy foods such as baked goods, chips, sugar-sweetened beverages and candy applied at the point-of-sale at the register (not reflected in shelf prices). Passed earlier in 2014 [13]), part of the HDNA also included a waiver of conventional sales tax (at that time 5%, currently 6%) on healthy foods including water, fresh fruits and vegetables and nuts [14]. While several large municipalities in the U.S. (such as Berkeley and Philadelphia [15,16]) have sweetened beverage taxes and international examples exist of unhealthy food taxes [17–21], the HDNA is the first-ever tax on unhealthy foods in the U.S. and any tribal or indigenous nation worldwide with a rural population at high-risk for common chronic diseases.

However, to date, no study has assessed changes in the store food environment after implementation of the HDNA legislation in 2014. Using a follow-up of a validated store survey originally implemented in 2013, this study assessed changes in pricing and availability of healthy and unhealthy foods across Navajo Nation stores from 2013 to 2019. Because the 2013 survey also sampled stores located in border towns adjacent to the Navajo Nation, we sought to compare trends over the past six years on and off the reservation in order to determine whether HDNA could be influencing changes, including increases or decreases in availability and pricing of healthy and less healthy foods.

## Methods

### Store selection

This was an observational study with repeated measurements compared across two time points: 2013 and 2019. In 2013, a survey called Epi-Aid was conducted to assess the Navajo food store environment [9]. The total Epi-Aid sample in 2013 included 83 stores, of which 63 were on the Navajo Nation (13 grocery, 50 convenience stores) and 20 stores in five border towns. Data on stores was selected from a national proprietary InfoUSA dataset of businesses [22], augmented with information from the Yellow Pages, Google Maps and local residents. Stores were selected if they were accessible via paved roads, and included both chain and independent grocery and convenience stores (including trading posts) and farmers' markets on the Navajo Nation [23].

The 2013 list was updated using the same databases and calling each chapter on the Navajo Nation within specified intervals (annually) to confirm existing and add new store information. This updated 2019 list was utilized to conduct a follow-up survey in 2019 among all stores from the 2013 study that were still operational. If original stores could not be surveyed, we used a computer-generated randomized algorithm to match the original store to an operational store based on store type (convenience store/grocery store), location (on/off reservation), and region/border town.

### Measures

General store information was recorded including geographic location, type of store (grocery or convenience), whether the store accepted Women Infants and Children (WIC) or Supplemental Nutrition Assistance Program (SNAP) benefits, presence of signage promoting healthy foods, unhealthy foods, and/or the HDNA law, and presence of a store snack shop or mini-concession stand. These were defined as additional venues located in a grocery or convenience store where on-the-go foods for immediate consumption such as corn dogs or nachos were sold.

The Nutrition Environment Measurement Survey-Stores (NEMS-S) [24,25] was used to assess food pricing and availability. The NEMS is a validated observation tool developed to assess the nutrition environment in community food and retail outlets including stores [24]. The instrument was adapted in 2013 to include commonly sold foods (including traditional foods such as blue corn meal, soft corn tortillas, mutton, squash). Additional items added to the original survey tool were adapted from the Nemours Healthy Vending Guide [26] to further facilitate assessing the availability of healthy and less healthy snacks within stores.

### Procedures

Surveyors participated in a formal day of in-person training led by one of the lead investigators, and included training on the NEMS-S store assessment survey and general information about troubleshooting. Procedures followed were the same for both the 2013 and 2019 observations to allow for comparisons. First, verbal approval to conduct store assessments and store manager surveys was obtained prior to any observation. Once approval was obtained, a surveyor entered the store and observed the food environment, including the signage, displays, foods offered and recorded pricing. Pricing was based on shelf price and did not include sales tax. Additional clarification was asked as needed (e.g., "does your store accept SNAP or WIC?").

A standard protocol included information on availability and pricing of major food categories of fresh fruits and vegetables, canned items, meat, beverages, water, flour/tortilla/bread,

chips and other snacks. Within each category, specific items were assessed (i.e. apples, bananas and oranges under fruits) and the number of items were recorded (ranging from 0 to 6 or more). If multiple brands of an item were available, the surveyor was asked to record the brand with the lowest price and record the size. Surveyors were asked to also find the lowest priced entrée item for a healthy and for a less healthy version, if foods for immediate consumption were available at the store. In the original 2013 Epi-Aid study, healthy or unhealthy classifications were based on the original NEMS or Nemours protocols [24,26], the 2010 Dietary Guidelines for Americans [27] or the USDA Nutrient database [28]. Based on this classification, the number of unhealthy and healthy foods were tabulated and an overall healthy to unhealthy ratio was calculated by dividing the number of foods in each group. All study procedures, measures were approved by the Navajo Nation Human Research Review board under protocol NNR-17.284.

## Analyses

Data were collected on printed surveys and then entered using Epi-Info software in 2013 [29]. In 2019, data were entered into an open-source application, CommCare on Samsung Galaxy[TM] Tablets (Suwon, Gyeonggi-do, South Korea). Study entry quality checks showed an error rate of less than 1%. Analyses included descriptive statistics and frequency distributions using RStudio Version 1.2 (RStudio Inc., Boston, MA) to characterize food availability, including the number and variety of healthy food options and traditional foods, pricing of healthful and unhealthful options (and ratio of the two) and acceptance of WIC or SNAP benefits (yes/no), promotion of local, organic or (un)healthy options (yes or no) and if the store demonstrated it implemented the HDNA legislation. These were calculated and averaged across all stores, and stratified by regional agency, type of store (convenience or grocery store) and location (on and off the Navajo Nation). Because traditional foods were only reported for the Navajo Nation stores in 2013, we report changes in traditional foods' availability from 2013 to 2019 only for the Navajo Nation stores. Pricing was adjusted for inflation to reflect changes in 2013 dollars using the Consumer Price Index as described by USAID [30]. For produce prices, grocery stores generally sold produce per pound and convenience stores per item. If a store offered both, we recorded the price per pound only as the cheapest price per unit. Since prices per pound and per piece cannot be directly compared, the changes from 2013 to 2019 are compared separately for cost per piece and per pound. Chi-square tests were used for hypothesis testing with categorical variables and t-tests for continuous variables to test whether food availability significantly changed from 2013 to 2019, using a p-value of 0.05. To assess significance of the difference-in-difference, changes in Navajo versus border stores over time, we built regression models, with time (2013 or 2019), location (Navajo Nation vs Border town), and an interaction term (Time*Location).

## Results

### Store sample

Of the original 83 stores from the Epi-Aid 2013 survey, 51 were successfully surveyed in 2019. The remaining 32 original stores could not be surveyed because they were no longer open (11), their store manager declined participation or needed corporate approval (15) or they could not be matched with the original database because either geocodes, region ID and/or store name were not available (6). Of the stores that could not be surveyed, a total of 88% of these stores were on the Navajo Nation (similar to the overall sample), 81% were convenience stores, 16% grocery stores and 3% other. For these 32 stores, we successfully matched 20 original stores to an operational store for 2019 sampling (see **Fig 1**). The total analytic sample

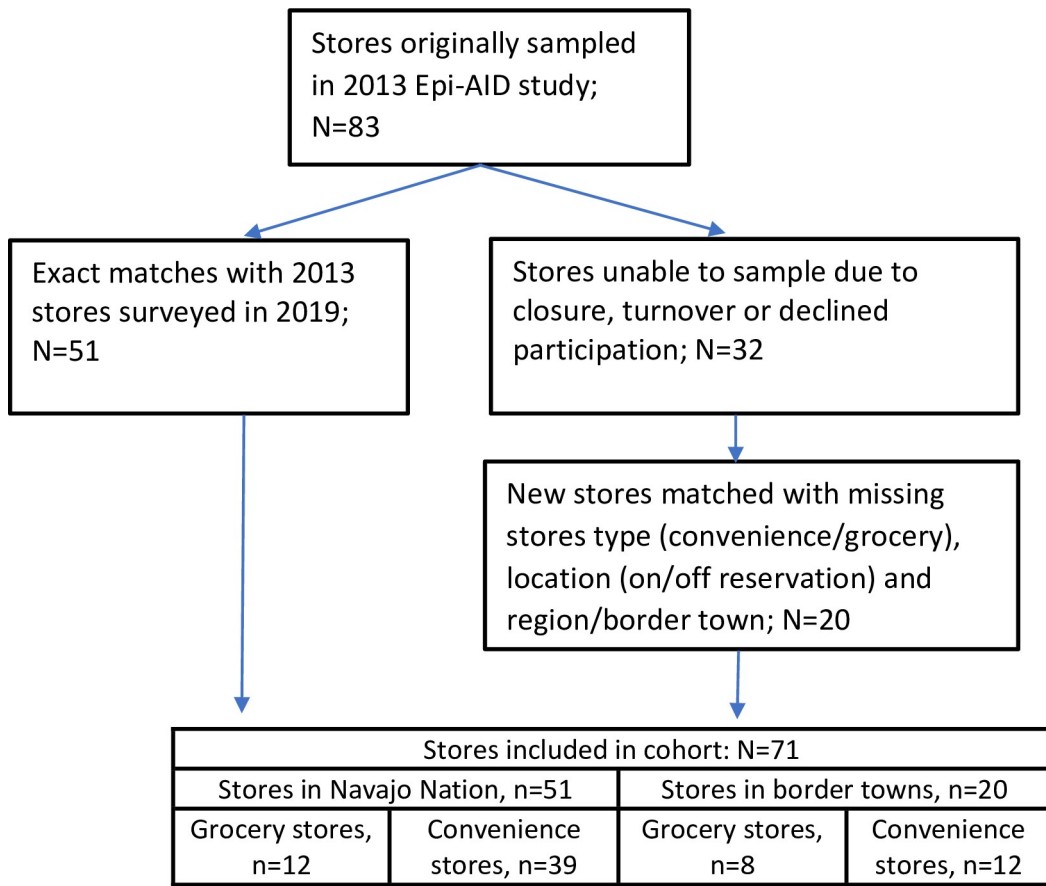

**Fig 1. Flowchart of store selection.**

included 71 stores, including 51 original stores and 20 matched pairs, representing all five regions of the Navajo Nation.

## Store characteristics

Of the 71 stores in the final analytic sample, a total of 72% of stores were located on the Navajo Nation (n = 51, of which 39 were convenience stores and 12 grocery stores) (see Table 1). Three quarters of all stores had additional venues for immediate or to-go consumption foods (snack shops or mini-concession stands) but only 16% of snack shops sold fruits or vegetables. Over half of stores (55%) accepted WIC and over three-quarters accepted SNAP in both 2013 and 2019. Grocery stores were significantly more likely to accept WIC (94%) compared with convenience stores (37%) (p = 0.001).

## Availability of healthier foods

In 2019, the vast majority of stores had fruit for sale, and more than half of stores had vegetables for sale. In 2013, Navajo stores were more likely to sell fresh produce compared with compared with stores located in border towns: 82% of Navajo stores v. 65% of border town stores sold fruits (p = 0.21) and 88% of Navajo stores v. 60% of border town stores sold vegetables (p = 0.018). Between 2013 and 2019, border town stores began increasing their offerings of fresh produce, such that there were no significant differences in produce availability among

**Table 1. Store characteristics of Navajo and border town stores, n = 71.**

|  | Matched Sample (n = 71) |
|---|---|
| **Store type** | N (%) |
| Convenience stores | 51 (72%) |
| Grocery stores | 20 (28%) |
| **Store Location** |  |
| Navajo Nation | 51 (72%) |
| Border town | 20 (28%) |
| **Region** |  |
| Southern | 7 (10%) |
| Eastern | 4 (6%) |
| Northern | 12 (17%) |
| Central | 5 (7%) |
| Western | 18 (25%) |
| Southwest | 5 (7%) |
| **Border town** |  |
| Gallup | 5 (7%) |
| Grants | 3 (4%) |
| Farmington | 5 (7%) |
| Flagstaff | 3 (4%) |
| Winslow | 4 (6%) |
| **Benefits** |  |
| Accepts WIC | 39 (55%) |
| Accepts SNAP | 57 (80%) |
| **Ready to eat options** |  |
| Snack shop | 53 (75%) |
| Salad Bar | 7 (10%) |

Navajo versus border town stores in 2019. A total of 82% of Navajo stores and 80% of border town stores sold fruit (p = 1.0), while 78% of Navajo stores and 70% of border town stores sold vegetables (p = 0.66). Among the 9 fresh produce items evaluated (see S1 Appendix), seven of these (apples, oranges, tomatoes, corn, celery, lettuce, and potatoes) were more frequently offered in Navajo stores than border town stores, although this comparison was statistically significant different for only one item, potatoes (p = 0.04).

These findings appear to be largely driven by trends happening in the border town convenience stores over time (see Table 2). In 2013, Navajo convenience stores were more likely to sell fresh fruits (p = 0.052) and fresh vegetables (p = 0.002) than border town convenience stores. By 2019, border town convenience stores had "caught up" to Navajo convenience stores in terms of fruit offerings. As for vegetables, even in 2019, the variety of fresh vegetables offered at Navajo convenience stores still exceeded those at border convenience stores; for example 62% of Navajo convenience stores offered at least three types of vegetables, compared with 17% of border town convenience stores (p = 0.017).

## Availability of traditional foods

From 2013 to 2019, the number of stores that offered any traditional food (blue corn, yellow corn, mutton, wild animals or wild plants) remained nearly constant at 29 stores in 2013 and 30 stores in 2019, though what stores offered changed over time. All Navajo grocery stores and 16 (41%) Navajo convenience stores sampled in 2019 sold traditional foods. As shown in

**Table 2. Percentage of convenience stores offering healthier items in 2013 and 2019, n = 51.**

| | 2013 | | | 2019 | | |
|---|---|---|---|---|---|---|
| | Navajo convenience stores (N = 39) n (%) | Border convenience stores (N = 8) n (%) | P-value* | Navajo convenience stores (N = 39) n (%) | Border convenience stores (N = 8) n (%) | P-value* |
| Any fruit | 30 (77%) | 5 (42%) | 0.052 | 30 (77%) | 8 (67%) | 0.738 |
| ≥3 types of fruit | 21 (54%) | 2 (17%) | 0.053 | 23 (59%) | 6 (50%) | 0.829 |
| Apples | 27 (69%) | 4 (33%) | 0.059 | 29 (74%) | 6 (50%) | 0.116 |
| Oranges | 25 (64%) | 2 (18%) | 0.018 | 26 (67%) | 5 (42%) | 0.141 |
| Bananas | 19 (50%) | 5 (42%) | 0.863 | 19 (49%) | 7 (58%) | 0.228 |
| Any vegetables | 33 (85%) | 4 (33%) | 0.002 | 28 (72%) | 6 (50%) | 0.294 |
| ≥3 types of vegetables | 25 (64%) | 4 (33%) | 0.121 | 24 (62%) | 2 (17%) | 0.017 |
| Tomatoes | 20 (51%) | 4 (33%) | 0.448 | 21 (54%) | 2 (17%) | 0.023 |
| Corn | 0 | 0 | | 3 (8%) | 0 | |
| Celery | 9 (23%) | 1 (8%) | 0.478 | 15 (39%) | 0 | 0.035 |
| Lettuce | 25 (64%) | 4 (33%) | 0.121 | 18 (46%) | 2 (17%) | 0.113 |
| Potatoes | 26 (67%) | 3 (25%) | 0.027 | 21 (54%) | 1 (8%) | 0.005 |
| Squash | 2 (5%) | 1 (8%) | 1 | 3 (8%) | 0 | 0.513 |

* P-values comparing convenience stores on the Navajo Nation to border towns.

**Fig 2**, the availability of mutton (p = 0.154), blue corn (p = 0.492) and wild plants and berries (p = 0.476) increased over time in both grocery and convenience stores and the availability of yellow corn (p = 0.067) decreased but none of these changes were significant. At both time points, grocery stores were more likely to carry traditional foods than convenience stores on the Navajo Nation (p<0.001).

## Pricing

The average cost per item of fruit was higher in 2013 on the Navajo Nation compared with those sold in border town stores ($0.88 vs $0.63, p = 0.06). From 2013 to 2019, after adjusting for inflation, the average cost per item of fruit decreased by 13% in stores on the Navajo Nation (from $0.88 to $0.76, p = 0.029- comparing average cost of fruit in 2013 vs 2019 in NN stores) and

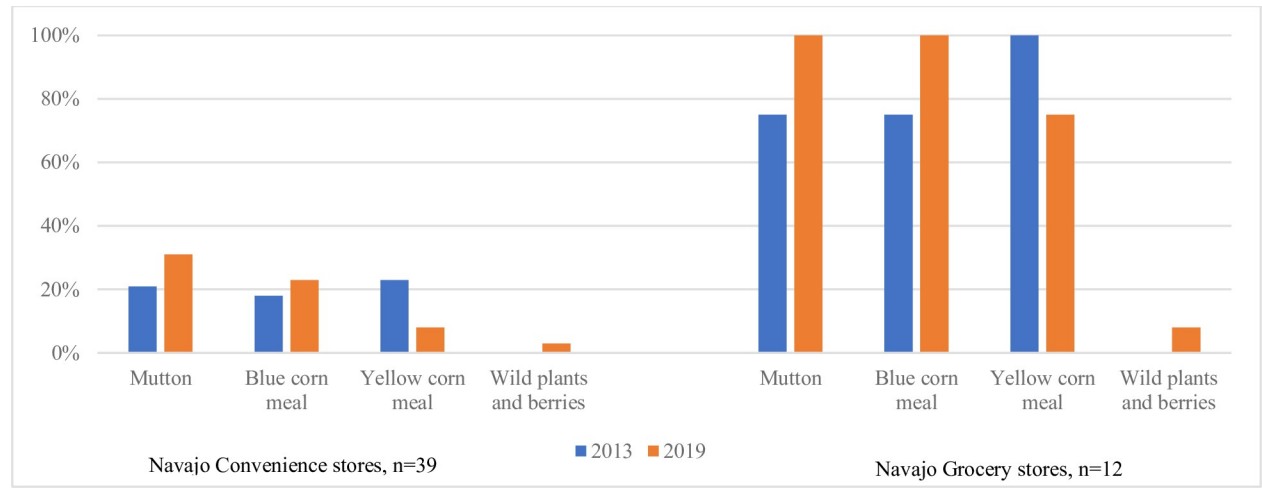

**Fig 2. Trends in traditional foods sold in Navajo stores by type, n = 51.**

**Table 3. Inflation-adjusted average price per item among convenience stores, n = 51.**

| | Average price per item, Navajo Nation stores (n = 39) | | | Average price per item, Border town stores (n = 12) | | |
|---|---|---|---|---|---|---|
| | **2013** | **2019** | **p-value*** | **2013** | **2019** | **p-value*** |
| **Fresh fruit** | | | | | | |
| Apples | $1.01 (n = 24) | $0.80 (n = 29) | 0.007 | $0.87 (n = 4) | $0.83 (n = 6) | 0.871 |
| Oranges | $0.90 (n = 23) | $0.83 (n = 25) | 0.574 | ** | $0.83 (n = 5) | N/A |
| Bananas | $0.72 (n = 22) | $0.59 (n = 21) | 0.306 | $0.45 (n = 5) | $0.58 (n = 7) | 0.505 |
| Average price | $0.88 (n = 69) | $0.76 (n = 75) | 0.029 | $0.63 (n = 9) | $0.73 (n = 18) | 0.519 |
| **Fresh vegetables** | | | | | | |
| Celery | $2.29 (n = 6) | $2.51 (n = 18) | 0.945 | ** | ** | N/A |
| Lettuce | $2.04 (n = 25) | $2.14 (n = 24) | 0.509 | $1.54 (n = 4) | $1.72 (n = 3) | 0.710 |
| Potato (per pound) | $0.51 (n = 19) | $0.58 (n = 18) | 0.493 | ** | $0.52 (n = 9) | N/A |

increased from $0.63 to $0.73 in border stores (p = 0.519- comparing average cost of fruit in 2013 versus 2019 in border stores) (see Table 3). As a result, the cost of fresh fruits on Navajo versus border stores were much more comparable in 2019, compared with 2013. While data on vegetable prices were sparse, they did not change significantly during this period in either region.

In terms of other foods and beverages (see S2 Appendix), the price of water was slightly lower in 2019 ($0.99 per gallon) compared to 2013 ($1.04 per gallon) in Navajo grocery stores (4.8% decrease) and $1.71 to $1.52 in Navajo convenience stores (11.1% decrease); however, border grocery and convenience stores saw similar declines. In 2013, the price of 100% juice was much higher in Navajo grocery stores ($3.92 per 59oz) relative to border grocery stores ($2.94 per 59oz), but much closer in 2019 ($3.04 and $2.87). During the same time, changes in pricing of chips, juice drinks and soda were inconsistent. For example, the price of a 10oz bag of Lays' chips increased in Navajo grocery stores from $3.23 to $3.63 but decreased in border grocery stores from $3.31 to $2.85, whereas the pattern was the opposite for convenience stores (decrease on the Navajo Nation, increase in border convenience stores).

## Health promotion

Among Navajo stores, signage promoting locally grown or organic items did not change, while the promotion of organic items in border town stores did increase over time (see Table 4). The vast majority of stores (>80%) did not identify or promote locally grown or organic items.

**Table 4. Changes in promotion and signage among Navajo and border town stores, n = 71.**

| | 2013 | | | 2019 | | | Change | | |
|---|---|---|---|---|---|---|---|---|---|
| | **Navajo stores (n = 51)** | **Border town stores (n = 20)** | **p-value** | **Navajo stores (n = 51)** | **Border town stores (n = 20)** | **p-value** | **Navajo stores** | **Border town stores** | **p-value** |
| Promotion of locally grown items | 18% | 30% | 0.410 | 14% | 25% | 0.431 | -4% | -5% | 0.959 |
| Promotion of organic items | 6% | 25% | 0.061 | 8% | 40% | 0.004 | 2% | 15% | 0.714 |
| Signage promoting healthy eating | 22% | 30% | 0.660 | 39% | 40% | 1.00 | 17% | 10% | 0.626 |
| Signage promoting less healthy eating | 28% | 40% | 0.457 | 46% | 60% | 0.427 | 18% | 20% | 0.999 |
| Signage promoting HDNA | 0% | 0% | N/A | 6% | 0 | 0.651 | 6% | N/A | N/A |

*P-values comparing stores on the Navajo Nation to border towns in 2013, 2019 and difference-in-difference/change over time.

Both on and off the reservation, signage promoting both healthy and less healthy eating increased. While not a requirement for any store, three stores on the Navajo Nation had promotional signs or other information (i.e. flyers) showing that they implemented the Healthy Diné Nation Act (HDNA) in 2019.

### Sensitivity analysis

A sensitivity analysis limited to only stores which were successfully measured in 2013 and 2019 (n = 51) yielded very similar findings to those of the full cohort (n = 71). Average absolute pricing difference between the exact matched sample and full cohort was 3.1 cents (some were higher and some lower, and the summed averaged difference was only 0.4 cents). The largest item difference in 2019 was for celery, which was $1.81 per pound in the full cohort and $1.93 in the exact matched sample, a $0.12 difference. The same comparisons were significant in the exact matched sample as in the full cohort.

## Discussion

This study presents the first assessment of changes in the food environment following the implementation of the Healthy Diné Nation Act of 2014 (HDNA), which included a 2% tax on unhealthy foods and waived 5% sales tax on healthy items. The HDNA was the first-ever such legislation among an indigenous or rural community at high risk for diabetes and other conditions. Among a broad sample of 51 stores on the Navajo Nation, findings reveal high overall availability of healthy items and modest improvements in the Navajo food store environment. Notably, 71% of Navajo stores offered 3 or more varieties of fruit and 65% sold 3 or more different vegetables. Promotion of healthy eating increased by 18%, and more stores offered traditional foods in 2019 compared with 2013. While not a requirement in the legislation, several stores had signage promoting the HDNA legislation itself, showing their desire to communicate or advertise the legislation to their consumers. In addition, in 2019 cost of fruits was about 17% lower than in 2013, with similar decreases in 100% juice and water.

Despite diet-related disparities in rural communities, very little research has focused on the food store environment in rural areas. For example, in a recent review of factors that influence food store manager decision making regarding healthy food promotion and environmental modification, only 7 of the 31 studies were in rural settings [31]. The review indicated that challenges in rural areas include a shrinking consumer base [32] and reduced demand for produce in summer due to gardening [33], while partnerships with local farmers were cited as a facilitator [34]. However, none of these studies reviewed food store environment changes following a policy like the HDNA. Our study included data from border town stores in order to explore whether any changes in the store environment were unique to the Navajo Nation. We did not identify major consistent differences in trends in pricing or availability between Navajo versus border towns. However, Navajo stores did see a larger decrease in pricing of fruit and 100% juice, bringing their pricing more in line with border towns. While changes cannot be directly linked to the HDNA, these price improvements, combined with greater availability of traditional foods, and increased health promotion signage reflect a trend toward healthier food environments in Navajo store over the six-year period.

Prior research has suggested that following taxation of unhealthy foods, consumption of unhealthy foods decreased, especially among low-income populations [17–21,35]. However, in Mexico, effects of sugar-sweetened beverage taxation were less pronounced in rural areas as the cost of the tax was not entirely passed on to the consumers, defined as a population of less than 2,500 [20–21], similar to the Navajo Nation where the average community size is 1,650 residents [1]. While compliance with HDNA legislation is not fully known at this time and

pricing changes may be inconsistent, HDNA revenue has decreased approximately 3% each year, suggesting lower consumption of unhealthy foods or shifting purchases of these items to border towns [35]. Further study is needed to assess whether improvements in the food store environment could moderate or amplify the influence of HDNA on purchasing behavior.

In comparing convenience store trends on the Navajo Nation with those in neighboring towns, some interesting findings emerged. In 2013, more Navajo convenience stores offered fresh produce compared with border town stores. By 2019, while border town convenience stores had reached comparable levels of fresh fruit availability, Navajo convenience stores still exceeded fresh produce availability than border convenience stores in 2019. These findings may reflect the important role of many small stores–including trading posts–on the Navajo Nation: access points for "non-snack" staple foods for the local community. In fact, in a 2019 survey of shoppers exiting small stores on the Navajo Nation, most consumers (72%) reported shopping at that store at least weekly and most (76%) also lived within 30 minutes from the store [36]. In light of the expansive rural geography and paucity of grocery stores on the Navajo Nation, the promising trends toward affordable produce and more traditional foods suggest that small stores could play an important role in reducing food insecurity on the reservation.

Pricing trends among vegetables and other healthy foods were inconsistent, and promotion of local or organic foods was still low. These findings suggest further opportunities for additional health promotion in the Navajo Nation food store environments. Research has already documented that the store managers are interested in providing healthier options [12], and interventions such as placing produce at the point-of-sale, providing culturally appropriate promotional materials, staff training on produce handling and reimbursement for fruit and vegetable vouchers for high-risk families merit further exploration [36–38].

## Limitations

The current study has several limitations. First, there are natural sample size limitations due to the number of stores on the Navajo Nation, limiting statistical power in several comparisons. For example, there are only 13 grocery stores on the Navajo Nation and they were all surveyed in both 2013 and 2019. Second, not all stores could be surveyed in 2019 and the matching approach that ensured 20 stores without a direct match were the same type, location (on/off reservation) and region/border town has limitations. However, the sensitivity analysis limited to only stores which were successfully measured in 2013 and 2019 (n = 51) yielded similar findings to those of the full cohort (n = 71). Third, pricing data was based on marked shelf prices, not purchases. While these procedures were the same as in 2013 and no shelf prices were missing, item cost can potentially be marked incorrectly and sales tax including the HDNA was not included in the price. Further, the 2013 and 2019 data present snapshots of the cost and availability of foods at one time-point, rather than documenting price fluctuations over time. However, data collection occurred in the same season in both 2013 and 2019, in an effort to limit the effect of any seasonal impact on pricing and availability. Finally, while increases in health promotion and direct signage about the HDNA suggest some increased awareness, the extent to which the HDNA legislation may have spurred any changes requires further study; in-depth interviews with store managers would be needed to gain further insight into underlying barriers and facilitators of healthy changes, HDNA implementation and promotion in the food store environment.

## Conclusions

This was the first study to assess changes in the Navajo Nation food store environment following implementation of the Navajo Nation legislation combining a 2% tax on unhealthy foods

with a waiver of 5% sales tax on healthy foods. Although not necessarily linked to HDNA, we observed improvements in pricing and varieties of healthy produce, primarily in Navajo convenience stores. Due to the low density of grocery stores, convenience stores play a major role in the food availability and modest improvements have the potential to impact food accessibility for rural tribal communities at high-risk for food insecurity and food-related chronic conditions.

## Supporting information

**S1 Appendix. Percentage of Navajo and border town stores offering healthier items in 2013 and 2019, n = 71.**
(DOCX)

**S2 Appendix. Inflation-adjusted average price of food items in Navajo and border town grocery and convenience stores, 2013 and 2019.**
(DOCX)

## Acknowledgments

We wish to thank the stores that have graciously participated in this study.

## Author Contributions

**Conceptualization:** Carmen George, Hendrik Dirk de Heer, Ramona Antone-Nez, Sonya Sunhi Shin.

**Data curation:** Carmen George, Carolyn Bancroft, Shine Krystal Salt, Cameron S. Curley, Caleigh Curley, Sonya Sunhi Shin.

**Formal analysis:** Carolyn Bancroft, Hendrik Dirk de Heer, Sonya Sunhi Shin.

**Funding acquisition:** Hendrik Dirk de Heer, Del Yazzie, Sonya Sunhi Shin.

**Investigation:** Carmen George, Caleigh Curley, Hendrik Dirk de Heer, Del Yazzie, Regina Eddie, Ramona Antone-Nez, Sonya Sunhi Shin.

**Methodology:** Carmen George, Carolyn Bancroft, Hendrik Dirk de Heer, Del Yazzie, Sonya Sunhi Shin.

**Project administration:** Carmen George, Shine Krystal Salt, Cameron S. Curley, Caleigh Curley, Hendrik Dirk de Heer, Regina Eddie, Sonya Sunhi Shin.

**Resources:** Cameron S. Curley, Caleigh Curley.

**Software:** Cameron S. Curley.

**Supervision:** Carmen George, Hendrik Dirk de Heer, Del Yazzie, Sonya Sunhi Shin.

**Visualization:** Carolyn Bancroft, Shine Krystal Salt, Hendrik Dirk de Heer, Sonya Sunhi Shin.

**Writing – original draft:** Caleigh Curley, Hendrik Dirk de Heer, Sonya Sunhi Shin.

**Writing – review & editing:** Carmen George, Carolyn Bancroft, Shine Krystal Salt, Cameron S. Curley, Del Yazzie, Regina Eddie, Ramona Antone-Nez, Sonya Sunhi Shin.

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
