## [Decision Letter · Decision Letter 0]

23 Apr 2021

PONE-D-21-08279

Changes in food pricing and availability on the Navajo Nation following a 2% tax on unhealthy foods: the Healthy Diné Nation Act of 2014

PLOS ONE

Dear Dr. Shin,

Thank you for submitting your manuscript to PLOS ONE. After careful consideration, we feel that it has merit but does not fully meet PLOS ONE’s publication criteria as it currently stands. Therefore, we invite you to submit a revised version of the manuscript that addresses the points raised during the review process.

Please focus in particular on the comments related to the gap identified in the introduction and including some methodologic details regarding the border towns and display of prices on the shelves.

We look forward to receiving your revised manuscript.

Kind regards,

Maya K. Vadiveloo

Academic Editor

PLOS ONE

Journal Requirements:

4. We note that Figure 2 in your submission contain map images which may be copyrighted. All PLOS content is published under the Creative Commons Attribution License (CC BY 4.0), which means that the manuscript, images, and Supporting Information files will be freely available online, and any third party is permitted to access, download, copy, distribute, and use these materials in any way, even commercially, with proper attribution. For these reasons, we cannot publish previously copyrighted maps or satellite images created using proprietary data, such as Google software (Google Maps, Street View, and Earth). For more information, see our copyright guidelines: http://journals.plos.org/plosone/s/licenses-and-copyright.

You may seek permission from the original copyright holder of Figure 2 to publish the content specifically under the CC BY 4.0 license. 

If you are unable to obtain permission from the original copyright holder to publish these figures under the CC BY 4.0 license or if the copyright holder’s requirements are incompatible with the CC BY 4.0 license, please either i) remove the figure or ii) supply a replacement figure that complies with the CC BY 4.0 license. Please check copyright information on all replacement figures and update the figure caption with source information. If applicable, please specify in the figure caption text when a figure is similar but not identical to the original image and is therefore for illustrative purposes only.

Reviewers' comments:

Reviewer's Responses to Questions

**Comments to the Author**

1. Is the manuscript technically sound, and do the data support the conclusions?

Reviewer #1: Yes

Reviewer #2: Yes

2. Has the statistical analysis been performed appropriately and rigorously? 

Reviewer #1: Yes

Reviewer #2: Yes

3. Have the authors made all data underlying the findings in their manuscript fully available?

Reviewer #1: No

Reviewer #2: No

4. Is the manuscript presented in an intelligible fashion and written in standard English?

Reviewer #1: Yes

Reviewer #2: Yes

5. Review Comments to the Author

Reviewer #1: Thank you for the opportunity to review this manuscript. As the authors note, no prior studies have assessed changes in the food store environment after the implementation of the HDNA, which is a topic of interest to health researchers and policymakers, among others. The paper is well-written and synthesizes existing research on the subject clearly.

My primary concerns relate to how price data was collected, particularly in convenience stores, in which prices may or may not be listed on the shelf. If they were all posted, then please specify that in the text, and if not, it would be helpful to know from what percentage of stores the prices were obtained from the store owner (or by purchasing the items). Additionally, it is unclear whether the sales tax was included in the prices, which is an important consideration given that this analysis looks at changes in price and the waiver of sales tax on the price of healthy foods is a key component of HDNA. Other, more minor suggestions are described below.

Ln 35: suggest “offered” in place of “sold” for clarity, since data was collected on product availability, and can’t tell us about sales.

Ln 56: “Tribal” is capitalized here, but not elsewhere (e.g., line 80). Suggest using whichever convention is preferred consistently throughout.

Ln 79: Philadelphia’s tax is technically a “sweetened beverage tax”, rather than a sugar-sweetened beverage tax since it applies to artificially sweetened beverages as well. Suggest choosing a different example city or dropping the “sugar-“.

Line 97: add “would” after “healthy foods”

Ln 98: What do you mean by “regional variability”? Did you stratify by region?

Ln 113: Sentence is a bit confusing- was the list updated in 2019 or was it updated annually?

Lns 145-150: Were prices listed on shelf tags (or the products themselves) or were prices obtained by asking the store owner/manager, or a combination? Did you collect data about whether prices were displayed in the store or not?

Also, if prices were listed on the shelf tag, was sales tax included, not included, or was it unknown? I think this is highly relevant given the content of the HDNA law.

Lns 161-164: What do you mean by the number/variety of healthy food options and the ratio of healthful/unhealthful options? Is this based on all items available in the store or a specific set of items from the NEMS tool?

Ln 164: How did stores demonstrate that they had implemented HDNA? Was this measured based on presence of a sign in the stores?

Lns 168-169: This is confusing. Are prices not presented unless the produce item was priced per item? Were they included in the analysis? This seems problematic if grocery stores typically price produce by weight.

Lns 180-181: Was drop-out/closure differential by location (on or off-reservation)? Also, what proportion of stores that dropped out (or closed) were grocery stores?

Ln 199: Sentence should make it clear that the p-value for fruit was non-significant (unless there is a typo).

Ln 210: add “more” before “likely.”

Ln 219: “Bordertown” sounds like a single place when capitalized and in the singular.

Lns 222-3: Is this out of all stores or out of Navajo stores only? Did no bordertown stores sell traditional foods?

Ln 287: “wasn’t” should be “was not.”

Ln 312: It is unclear who “they” are- is this store owners?

Table 2: Add total “Ns” in the header row (e.g., Navajo convenience stores, N=39)

Table 4: Suggest changing the title from “Trends” to “Changes”, since data is from two time points. Also, was there a reason p-values weren’t included for the Navajo store v. border store comparison?

Reviewer #2: Overall thoughts and summary:

The authors’ work is important and novel and merits publication with minor revisions.

Major comment:

• Overall, I was confused about how the authors conceptualized the link between the HDNA legislation and changes in pricing and availability of healthy and unhealthy foods in stores subject to the legislation. My impression after reading the Introduction was that the authors were exploring a potential mechanism for their previous work -- specifically that the food tax likely lead to a decrease in purchasing of unhealthy foods. However, lines 282-283 and 329-331 in the Discussion make me think otherwise. Why do the authors state that “it is not possible to determine whether changes in the food environment were due to the HDNA legislation or other factors”? If the rationale for looking at changes in pricing/availability are NOT related to the food tax, then why make it the focus of the introduction?

Minor comments:

• Can you state earlier, in the methods section, why you are comparing outcomes of interest between the Navajo nation and border towns?

• Line 233: Can you first state that the cost of fresh fruits was higher in Navajo (vs. border) stores at baseline?

• Lines 255-256 and 269-270: Can you comment further on the implications of so few stores demonstrating implementation of the HDNA legislation?

• Lines 324-326: Suggest adding this sensitivity analysis to the methods and results.

6. PLOS authors have the option to publish the peer review history of their article (what does this mean?). If published, this will include your full peer review and any attached files.

Reviewer #1: No

Reviewer #2: No

---

## [Author Response · Author response to Decision Letter 0]

21 Jun 2021

Article title: Changes in food pricing and availability on the Navajo Nation following a 2% tax on unhealthy foods: the Healthy Diné Nation Act of 2014

Responses to reviewer/editorial comments.

Editor comments:

Thank you for submitting your manuscript to PLOS ONE. After careful consideration, we feel that it has merit but does not fully meet PLOS ONE’s publication criteria as it currently stands. Therefore, we invite you to submit a revised version of the manuscript that addresses the points raised during the review process.

Please focus in particular on the comments related to the gap identified in the introduction and including some methodologic details regarding the border towns and display of prices on the shelves.

Overall Response: 

Thank you for these comments and thoughtful review of the manuscript overall. We feel it has been greatly strengthened as a result of the feedback. Notably, we have adjusted the introduction to more appropriately focus on changes in the Navajo Nation food environment and comparison to border town stores added clarification throughout to several methodological issues (and also added these as limitations in the discussion as appropriate). In addition, we have included how pricing data was collected, clarified issues regarding item pricing per piece and per pound, included additional details about the NEMS survey categories and clarified store promotion of the HDNA legislation. Detailed responses to reviewer comments are outlined below.

Reviewer comments:

Reviewer #1: Thank you for the opportunity to review this manuscript. As the authors note, no prior studies have assessed changes in the food store environment after the implementation of the HDNA, which is a topic of interest to health researchers and policymakers, among others. The paper is well-written and synthesizes existing research on the subject clearly.

My primary concerns relate to how price data was collected, particularly in convenience stores, in which prices may or may not be listed on the shelf. If they were all posted, then please specify that in the text, and if not, it would be helpful to know from what percentage of stores the prices were obtained from the store owner (or by purchasing the items). 

Response: Thank you for this comment. We have clarified that data collection was based on observational surveys and marked prices, not purchases. If pricing information was missing, store staff was inquired. The primary reason for this approach ¬¬was to follow the same procedures as were implemented as in 2013 to allow for direct comparison in pricing between 2013 and 2019. We have clarified this in the text and also added this as a limitation in the discussion section. 

Comment: Additionally, it is unclear whether the sales tax was included in the prices, which is an important consideration given that this analysis looks at changes in price and the waiver of sales tax on the price of healthy foods is a key component of HDNA. Other, more minor suggestions are described below.

Response: Sales tax was not included since the price information was based on marked prices. This has now also been clarified and added as a limitation to the discussion section.

Comment: Ln 35: suggest “offered” in place of “sold” for clarity, since data was collected on product availability, and can’t tell us about sales.

Response: This change was made.

Comment: Ln 56: “Tribal” is capitalized here, but not elsewhere (e.g., line 80). Suggest using whichever convention is preferred consistently throughout.

Response: We made it lowercase to be consistent throughout. 

Comment: Ln 79: Philadelphia’s tax is technically a “sweetened beverage tax”, rather than a sugar-sweetened beverage tax since it applies to artificially sweetened beverages as well. Suggest choosing a different example city or dropping the “sugar-“.

Response: Thank you! We dropped the word sugar. 

Comment: Line 97: add “would” after “healthy foods”

Response: This change was made. However, we subsequently removed the sentences as we want to be cognizant we are not analyzing sales data.

Comment: Ln 98: What do you mean by “regional variability”? Did you stratify by region?

Response: Apologies this was indeed unclear. We intended to compare the on-reservation to border town changes and have removed the text referring to regional variability (there are 5 regions on the Navajo Nation, but since the HDNA was applied across the entire Navajo Nation, so we did not compare HDNA implementation across Navajo Nation regions).

Comment: Ln 113: Sentence is a bit confusing- was the list updated in 2019 or was it updated annually?

Response: Apologies, this was confusing. The list was updated annually as part of general research team maintenance of the store database. The 2019 update was the list used for the follow-up data collection, which is why we referred to it specifically. We tried to clarify the sentence.

Comment: Lns 145-150: Were prices listed on shelf tags (or the products themselves) or were prices obtained by asking the store owner/manager, or a combination? Did you collect data about whether prices were displayed in the store or not? Also, if prices were listed on the shelf tag, was sales tax included, not included, or was it unknown? I think this is highly relevant given the content of the HDNA law.

Response: This is indeed important context. Prices were based on observational data and shelf prices using the NEMS survey to match the procedures implemented in 2013. Store staff was asked if pricing information was missing (the frequency was not formally recorded, but it was rare). We have aimed to clarify this throughout the manuscript and added this as a limitation.

Comment: Lns 161-164: What do you mean by the number/variety of healthy food options and the ratio of healthful/unhealthful options? Is this based on all items available in the store or a specific set of items from the NEMS tool?

Response: The number and variety of items was indeed based on the NEMS tool, which has 10 sections such as milk, breads & tortillas, fruits & vegetables, meats and hot dogs, canned foods etc. There are subsections under each section for specific individual types of foods; for example, 3 types of fruits and 6 types of vegetables are specifically listed and checked: apples, bananas, oranges, tomatoes, celery, lettuce, corn, squash, potatoes. It was also assessed if healthier versions of certain foods were available (i.e. low fat milk, lean ground beef, 100% juice). In addition, the tool specifies the number of items under each section (for example, for fruits, vegetables, chips etc.), with answer options being 0,1,2,3,4,5 and 6 or more. The data was summarized for the individual foods and the overall number of foods available under each section. Finally, foods were categorized as healthful or unhealthful and the ratio of these was calculated by dividing the total number of available food types in each category. We have added more detail to the methods section to clarify the procedures and ratio.

Comment: Ln 164: How did stores demonstrate that they had implemented HDNA? Was this measured based on presence of a sign in the stores?

Response: Indeed, this was noted based on presence of a sign or other promotional information (flyer etc.) in the store. This has been added to clarify.

Comment: Lns 168-169: This is confusing. Are prices not presented unless the produce item was priced per item? Were they included in the analysis? This seems problematic if grocery stores typically price produce by weight.

Response: Apologies, we agree this was confusing. The NEMS survey indicates to record the price per cheapest unit being sold. If a store offered a type of produce both per-piece and per-pound, only the per-pound price was recorded as it almost always is the cheapest per unit price. Because it is challenging to aggregate data across per-piece or per-pound pricing methods, comparisons between 2013 and 2019 pricing were made separately for per piece and per pound pricing. Indeed, grocery stores typically only sold produce per pound. For example, in the full cohort, 17 stores in 2013 offered apples only per pound and 18 in 2019, and the average price was compared between these stores for the cost of apples per pound. We have attempted to clarify the abovementioned sentences to reflect this approach more clearly. 

Comment: Lns 180-181: Was drop-out/closure differential by location (on or off-reservation)? Also, what proportion of stores that dropped out (or closed) were grocery stores?

Response: Of the 32 stores that were closed/declined participation or had missing geocode data, 88% were on the Navajo Nation, 12% in border towns, 81% were convenience stores, 16% grocery stores and 3% other. This information has been added to the manuscript. 

Comment: Ln 199: Sentence should make it clear that the p-value for fruit was non-significant (unless there is a typo).

Response: We added the letters ns. to indicate the p-value was indeed not significant. This was incidentally not a typo, as the p-value was 1.0 rounded up as percentages were nearly identical for convenience stores offering fruits on the Navajo Nation (82%) and in bordertowns (80%). We have also added .ns for the other comparisons in the same paragraph that were not statistically significant.

Comment: Ln 210: add “more” before “likely.”

Response: Thank you, this change was made.

Comment: Ln 219: “Bordertown” sounds like a single place when capitalized and in the singular.

Response: We agree and revised this to read: “P-value comparing convenience stores on the Navajo Nation to border towns.”

Comment: Lns 222-3: Is this out of all stores or out of Navajo stores only? Did no bordertown stores sell traditional foods?

Response: Unfortunately traditional foods were only assessed in the Navajo stores in 2013. Thus, we only reported traditional food availability in the Navajo grocery and convenience stores in 2019 as well to facilitate comparability between samples. We have added this clarification to the methods section.

Comment: Ln 287: “wasn’t” should be “was not.”

Response: This change was made.

Comment: Ln 312: It is unclear who “they” are- is this store owners?

Response: Indeed, these are the store managers. This was changed.

Comment: Table 2: Add total “Ns” in the header row (e.g., Navajo convenience stores, N=39)

Response: The N’s were added to the header row.

Comment: Table 4: Suggest changing the title from “Trends” to “Changes”, since data is from two time points. Also, was there a reason p-values weren’t included for the Navajo store v. border store comparison?

Response: We agree and Trends was changed to Changes. P-values were added to Table 4. 

Reviewer #2: Overall thoughts and summary:

The authors’ work is important and novel and merits publication with minor revisions.

Major comment:

• Overall, I was confused about how the authors conceptualized the link between the HDNA legislation and changes in pricing and availability of healthy and unhealthy foods in stores subject to the legislation. My impression after reading the Introduction was that the authors were exploring a potential mechanism for their previous work -- specifically that the food tax likely lead to a decrease in purchasing of unhealthy foods. However, lines 282-283 and 329-331 in the Discussion make me think otherwise. Why do the authors state that “it is not possible to determine whether changes in the food environment were due to the HDNA legislation or other factors”? If the rationale for looking at changes in pricing/availability are NOT related to the food tax, then why make it the focus of the introduction?

Response: We agree this was confusing and we have changed the introduction to limit the focus on any reductions in purchasing of unhealthy foods since purchase data are not available, and instead emphasize the focus on changes in the Navajo food store environment between 2013 and 2019 and compared to border towns. Lines 282-283 were removed and lines 329-331 rephrased to indicate that modest improvements (i.e. Navajo store pricing ‘catching up’ with border stores in some cases like fruits and 100% juice, more health promotion, explicit HDNA signage etc.) suggest some extent of increased awareness, but that further study is needed to which extent these were spurred by the HDNA. In-depth interviews with store managers will take place throughout 2021 to gain further insight into these questions.

Minor comments:

Comment: • Can you state earlier, in the methods section, why you are comparing outcomes of interest between the Navajo nation and border towns?

Response: We have now stated this earlier. Our main reason was that border towns are not subject to the HDNA legislation (neither the waiver of sales tax on healthy foods nor the added 2% HDNA tax on unhealthy foods. In addition, food accessibility issues on the Navajo Nation are a key reason behind the work in general and Navajo stores being able to offer similar healthy food options at similar price points to the border towns would be an important step in improving the food environment for the Navajo people.

Comment: • Line 233: Can you first state that the cost of fresh fruits was higher in Navajo (vs. border) stores at baseline?

Response: We have now added this.

Comment: • Lines 255-256 and 269-270: Can you comment further on the implications of so few stores demonstrating implementation of the HDNA legislation?

Response: We have now added language clarifying that these were stores that had promotional signage explicitly showing they were implementing the HDNA. There is no requirement for the stores to advertise the HDNA implementation. As a result, it is possible that store managers simply did not prioritize adding signage or did not want to necessarily advertise they were taxing an additional 2% on unhealthy foods. We are currently conducting interviews with store managers to gain insight into questions such as these about communication of the HDNA to consumers and motivation behind adding promotional signage about the HDNA. We have added this in the text to clarify.

Comment: • Lines 324-326: Suggest adding this sensitivity analysis to the methods and results.

Response: The sensitivity analysis was added to the methods and results section. Notably, results were highly similar: on average, prices were about 3 cents different between the exact matched sample and total sample (some were higher and some lower, when all values were added, average summed difference was 0.4 cents per item). The largest difference for any one item in 2019 was 12 cents for celery per pound ($1.81 in the total cohort, $1.93 in the exact matched sample). The same comparisons were still statistically significant.

---

## [Decision Letter · Decision Letter 1]

28 Jul 2021

PONE-D-21-08279R1

Changes in food pricing and availability on the Navajo Nation following a 2% tax on unhealthy foods: the Healthy Diné Nation Act of 2014

PLOS ONE

Dear Dr. Shin,

Thank you for submitting your manuscript to PLOS ONE. After careful consideration, we feel that it has merit but does not fully meet PLOS ONE’s publication criteria as it currently stands. Therefore, we invite you to submit a revised version of the manuscript that addresses the points raised during the review process.

We look forward to receiving your revised manuscript.

Kind regards,

Maya K. Vadiveloo

Academic Editor

PLOS ONE

Journal Requirements:

Additional Editor Comments (if provided):

Reviewers' comments:

Reviewer's Responses to Questions

**Comments to the Author**

1. If the authors have adequately addressed your comments raised in a previous round of review and you feel that this manuscript is now acceptable for publication, you may indicate that here to bypass the “Comments to the Author” section, enter your conflict of interest statement in the “Confidential to Editor” section, and submit your "Accept" recommendation.

Reviewer #1: (No Response)

Reviewer #2: All comments have been addressed

2. Is the manuscript technically sound, and do the data support the conclusions?

Reviewer #1: Yes

Reviewer #2: Yes

3. Has the statistical analysis been performed appropriately and rigorously? 

Reviewer #1: Yes

Reviewer #2: Yes

4. Have the authors made all data underlying the findings in their manuscript fully available?

Reviewer #1: Yes

Reviewer #2: Yes

5. Is the manuscript presented in an intelligible fashion and written in standard English?

Reviewer #1: Yes

Reviewer #2: Yes

6. Review Comments to the Author

Reviewer #1: Overall

This is interesting and well-written paper on a topic that has not been published about previously. The authors’ revisions have improved the clarity of the paper, although there are still some remaining issues to be addressed. Most of my suggestions are minor, although there is one methodological question to be considered.

Comments:

Line 30: The word “of” is missing after “the majority.”

Line 35: I suggest revising to say “While more Navajo stores offered traditional foods in 2019 (54%) compared to 2013 (38%), this trend was not statistically significant” to make the comparison clearer.

Lines 73-77: My understanding is that the HDNA is a sales tax, which is applied at the point of purchase. I suggest adding an additional sentence or clause to clarify that the prices of items subject to the HDNA likely do not reflect the price increase due to the tax, which would be applied at the register (if that is correct).

Line 122: Suggest deleting the word “popular.”

Lines 169-171: Most of the comparisons you make are between the Navajo Nation stores and the border town stores at the same time point. I don’t see a problem with that; however, you do in some cases compare the difference in availability or price over time for Navajo Nation stores compared to border town stores in the text (e.g., the abstract & lines 233-235). In most cases, you don’t report on the statistical significance of the change (“The cost of fresh fruit decreased by 18% in Navajo stores, compared with 6% in border stores”), which you may want to consider doing, so that the reader knows if the change is in fact significantly different from zero, and if the change in Navajo stores is statistically significantly different from the change in border stores (difference-in-differences).

Additionally, you report in the abstract that “While more Navajo stores offered traditional foods (38% in 2013 v. 54% in 2019), this trend was not statistically significant.” The section in the paper on traditional food availability does not include statistical tests, so it is not clear how this was compared. Given that you have matched samples, if you are comparing the same group of stores (e.g., Navajo Nation stores) at two time points, I believe you should use a paired t-test for continuous variables (or a non-parametric test like the Wilcoxon signed rank test if the change in price for the item is not normally distributed) and McNemar’s test for binary variables.

Line 179: missing “a” before “total”

Line 189: Suggest adding “all” before “stores” so that it is clear that you are talking about the full sample, not only the Navajo Nations stores. Also, 75% is substantially more than half, so you might re-word to say “Most stores (75%) had additional venues...” or something similar.

Lines 233-235: These lines currently refer the reader to Table 3, which does not show any measure of statistical significance, nor does it display the average cost of fresh fruit overall or the difference in average cost (referenced in the abstract and the lines mentioned).

Table 4 shows the statistical significance of the difference between sign presence in Navajo stores and border towns in 2013 and again in 2019, with a column showing the magnitude of the change over time by store location (on or off reservation), but no statistical measures. The table is also missing a footnote describing what the p-values indicate.

Line 279: The table indicates a difference of 17% for signs promoting healthy eating, not 18%.

Line 299: If prices were measured in the same season in 2013 and 2019, shouldn’t this be a six-year period?

Line 312: Suggest checking for consistency throughout the paper for use of “Navajo Nation” v. “the Navajo Nation.”

Line 321: There is a typo here (“Nation Nation”).

Line 343: Suggest adding “including the HDNA” after “sales tax”, since the HDNA is also a type of sales tax (if this is accurate).

Line 356: Suggest deleting “likely” before “necessarily.”

Line 360: Unclear what high-risk refers to – food security? Diet-related chronic disease?

Reviewer #2: (No Response)

7. PLOS authors have the option to publish the peer review history of their article (what does this mean?). If published, this will include your full peer review and any attached files.

Reviewer #1: No

Reviewer #2: No

---

## [Author Response · Author response to Decision Letter 1]

10 Aug 2021

Article title: Changes in food pricing and availability on the Navajo Nation following a 2% tax on unhealthy foods: the Healthy Diné Nation Act of 2014

Responses to reviewer/editorial comments.

Editor comments:

Thank you for submitting your manuscript to PLOS ONE. After careful consideration, we feel that it has merit but does not fully meet PLOS ONE’s publication criteria as it currently stands. Therefore, we invite you to submit a revised version of the manuscript that addresses the points raised during the review process.

Response: Thank you for the opportunity to further clarify the manuscript and add additional information on statistical significance. We have made the requested revisions and outlined responses to each comment below.

Reviewer #1: Overall

Comment: This is interesting and well-written paper on a topic that has not been published about previously. The authors’ revisions have improved the clarity of the paper, although there are still some remaining issues to be addressed. Most of my suggestions are minor, although there is one methodological question to be considered.

Response: Thank you for your thoughtful and detailed review. We have appreciated the opportunity for additional clarification and believe the manuscript was further refined and improved as a result.

Comments:

Line 30: The word “of” is missing after “the majority.”

Line 35: I suggest revising to say “While more Navajo stores offered traditional foods in 2019 (54%) compared to 2013 (38%), this trend was not statistically significant” to make the comparison clearer.

Response: We agree and have made these revisions.

Comment: Lines 73-77: My understanding is that the HDNA is a sales tax, which is applied at the point of purchase. I suggest adding an additional sentence or clause to clarify that the prices of items subject to the HDNA likely do not reflect the price increase due to the tax, which would be applied at the register (if that is correct).

Response: Indeed, this is correct and a good point of clarification. We have added a sentence to clarify the HDNA tax was applied at the point-of-sale at the register.

Comment: Line 122: Suggest deleting the word “popular.”

Response: The word popular has been removed.

Comment: Lines 169-171: Most of the comparisons you make are between the Navajo Nation stores and the border town stores at the same time point. I don’t see a problem with that; however, you do in some cases compare the difference in availability or price over time for Navajo Nation stores compared to border town stores in the text (e.g., the abstract & lines 233-235). In most cases, you don’t report on the statistical significance of the change (“The cost of fresh fruit decreased by 18% in Navajo stores, compared with 6% in border stores”), which you may want to consider doing, so that the reader knows if the change is in fact significantly different from zero, and if the change in Navajo stores is statistically significantly different from the change in border stores (difference-in-differences).

Additionally, you report in the abstract that “While more Navajo stores offered traditional foods (38% in 2013 v. 54% in 2019), this trend was not statistically significant.” The section in the paper on traditional food availability does not include statistical tests, so it is not clear how this was compared. Given that you have matched samples, if you are comparing the same group of stores (e.g., Navajo Nation stores) at two time points, I believe you should use a paired t-test for continuous variables (or a non-parametric test like the Wilcoxon signed rank test if the change in price for the item is not normally distributed) and McNemar’s test for binary variables.

Response: Thank you for these comments as we feel these analyses add valuable detail to the manuscript regarding statistical significance. For the pricing data, we updated the estimates for change in price and added p-values from the paired t-tests for both Navajo Nation stores and border towns. The initial comparisons averaged price changes for each fruit individually and then averaged the totals (initially, no paired comparison test was conducted). However, this did not take into account differences in sample sizes for individual fruits, so in the updated analyses, we calculated average change in price for all fruits combined (n=69 and n=75 for Navajo stores, n=9 and n=18 for border stores) and conducted t-tests. The updated analyses show a decrease of 13% in overall cost of fruit in Navajo Nation stores, from $0.88 to $0.76, which was significant p=0.029. The average cost in border town stores increased from $0.63 to $0.73, p=.519. Text in the results section and abstract were updated to reflect the changes. We also reformatted Table 3 to include the p-values for the comparison of change in average price between 2013 and 2019 for individual fruits and vegetables and also for average price per item of fruit. 

Regarding traditional food availability, we added further detail to provide additional description of the types of traditional foods (blue corn, yellow corn, mutton or wild animals or wild plants) to align with what was presented in Figure 2. The availability of mutton, blue corn and wild plants increased from 2013 to 2019, although changes were not statistically significant. We have replaced the original text with these more detailed analyses and added the p-values for tests of significance for each individual traditional food to the section on traditional foods. 

Comment: Line 179: missing “a” before “total”

Response: The ‘a’ was added..

Comment: Line 189: Suggest adding “all” before “stores” so that it is clear that you are talking about the full sample, not only the Navajo Nations stores. Also, 75% is substantially more than half, so you might re-word to say “Most stores (75%) had additional venues...” or something similar.

Response: We agree and this was changed to ‘Three quarters’ of all stores…”

Comment: Lines 233-235: These lines currently refer the reader to Table 3, which does not show any measure of statistical significance, nor does it display the average cost of fresh fruit overall or the difference in average cost (referenced in the abstract and the lines mentioned).

Response: We added the average cost of fresh fruit to the table, updated the abstract and text with this information and added p-values for the specific comparisons mentioned. None of the individual price findings were statistically significant, although the overall cost of fruit decreased significantly in Navajo Nation stores. 

Comment: Table 4 shows the statistical significance of the difference between sign presence in Navajo stores and border towns in 2013 and again in 2019, with a column showing the magnitude of the change over time by store location (on or off reservation), but no statistical measures. The table is also missing a footnote describing what the p-values indicate.

Response: We conducted a regression analysis to calculate the difference-in-difference, including a time x location interaction term. The description of the regression analysis was added to the methods section and the p-values for the difference in difference and footnote were added to Table 4. 

Comment: Line 279: The table indicates a difference of 17% for signs promoting healthy eating, not 18%.

Response: This was changed.

Comment: Line 299: If prices were measured in the same season in 2013 and 2019, shouldn’t this be a six-year period?

Response: This is correct and this was changed to a six-year period.

Comment: Line 312: Suggest checking for consistency throughout the paper for use of “Navajo Nation” v. “the Navajo Nation.”

Response: We have changed throughout the read the Navajo Nation.

Comment: Line 321: There is a typo here (“Nation Nation”).

Response: Thank you, this was changed.

Comment: Line 343: Suggest adding “including the HDNA” after “sales tax”, since the HDNA is also a type of sales tax (if this is accurate).

Response: Yes, this is correct and was added.

Comment: Line 356: Suggest deleting “likely” before “necessarily.”

Response: This was changed.

Comment: Line 360: Unclear what high-risk refers to – food security? Diet-related chronic disease?

Response: We revised the sentence to indicate that the high risk referred to food insecurity and food-related chronic conditions.

---

## [Editor Report · Decision Letter 2]

13 Aug 2021

Changes in food pricing and availability on the Navajo Nation following a 2% tax on unhealthy foods: the Healthy Diné Nation Act of 2014

PONE-D-21-08279R2

Dear Dr. Shin,

We’re pleased to inform you that your manuscript has been judged scientifically suitable for publication and will be formally accepted for publication once it meets all outstanding technical requirements.

Kind regards,

Maya K. Vadiveloo

Academic Editor

PLOS ONE

Additional Editor Comments (optional):

Thank you for your prompt and thoughtful responses to the reviewer feedback.
---

## [Editor Report · Acceptance letter]

25 Aug 2021

PONE-D-21-08279R2 

Changes in food pricing and availability on the Navajo Nation following a 2% tax on unhealthy foods: the Healthy Diné Nation Act of 2014 

Dear Dr. Shin:

I'm pleased to inform you that your manuscript has been deemed suitable for publication in PLOS ONE. Congratulations! Your manuscript is now with our production department. 

Kind regards, 

on behalf of

Dr. Maya K. Vadiveloo 

Academic Editor

PLOS ONE